

# Characterizing and correcting the warm bias observed in AMDAR temperature observations

Siebren de Haan[1], Paul M. A. de Jong[1], and Jitze van der Meulen[1]

[1]Royal Netherlands Meteorological Institute(KNMI), Wilhelminalaan 10, De Bilt 3732 GK, The Netherlands

**Correspondence:** Siebren de Haan (siebren.de.haan@knmi.nl)

**Abstract.** Some aircraft temperature observations, retrieved through the Aircraft Meteorological Data Relay (AMDAR), suffer from a significant warm bias when comparing observations with numerical weather prediction (NWP) model. In this manuscript we show that this warm bias of AMDAR temperature can be characterized and consequently reduced substantially. The characterization of this warm bias is based on the methodology of measuring temperature with a moving sensor and can be split

into two separate processes.

The first process depends on the flight phase of the aircraft and relates to difference of timing, as it appears that the time of measurement of altitude and temperature differ. When an aircraft is ascending or descending this will result in small bias in temperature due to the (on average) presence of an atmospheric temperature lapse rate.

The second process is related to internal corrections applied to pressure altitude without feedback to temperature observation

measurement.

Based on NWP model temperature data combined with additional information on Mach number and true airspeed, we were able to estimate corrections using data over an 18 months period from January 2017 to July 2018. Next, the corrections were applied on AMDAR observations over the period from September 2018 to mid-December 2019. Comparing these corrected temperatures with (independent) radiosonde temperature observations demonstrates a reduction of the temperature bias from

0.5K to around zero and reduction of standard deviation of almost 10%.

## 1 Introduction

Upper air observations from aircraft are an important source of information for numerical weather prediction (NWP). Amongst other sources, aircraft observations of temperature and wind are used to estimate the atmospheric state in order to initialize an NWP forecast run. Knowledge about the error characteristics is crucial for correct interpretation of the observation. The

presence of biases, which are persistent constant differences between observation and model, are detrimental for NWP performance (Dee and Da Silva, 1998). ECMWF has introduced an aircraft and flight phase dependent temperature correction (Cardinali et al., 2004). The so-called variational bias correction method (Dee, 2005) has been developed to remove the bias during assimilation, but the origin of the bias was not resolved.

In this manuscript the signature of the temperature bias from aircraft observations retrieved through the Aircraft Meteo-

rological Data Relay (AMDAR) is investigated. These error characteristics of AMDAR temperature observations have been





examined in a number of studies. A warm bias has been reported by Ballish and Kumar (2008). Drüe et al. (2007) observed aircraft type dependent systematic temperature errors. In general, the standard deviation of AMDAR minus radiosonde or AMDAR observations very close to each other is around 0.6K (Schwartz and Benjamin (1995), Benjamin et al. (1999)). The formal difference is slightly smaller than 0.4K (Painting, 2003).

In this manuscript the temperature bias is characterized, by assuming that the observed total bias consists of a flight phase part and a combined true airspeed and pressure related part. Information available through the Mode-S Enhanced Surveillance system (EHS) is combined to characterize and quantize each dynamic part. The Mode-S EHS system on its own can be used to derive temperature as described in de Haan (2011). These observations of temperatures are not exploited here, instead the Mach-number information and pressure altitude rate measured by an aircraft are used in the characterization. Temperatures

from NWP are used to calibrate for each aircraft individually the bias as a function of true airspeed and pressure.

This manuscript is organised as follows: first we discuss briefly the aircraft sensors used and the origin of the data used to determine pressure and temperature, Mach and true airspeed. Next we present the (possible) sources of temperature biases, and develop a methodology to quantify these biases. This section is followed by the description of the data preparation steps. The results are presented in Section 5, which is followed by the conclusions and discussion.

## 40   2   Aircraft sensors

For flight control and aircraft management, modern aircraft are equipped with sensors which are used to derive basic meteorological parameters. A pitot-probe measures static and total air pressure and an immersion thermometer probe is installed for total air temperature measurements. The information is sent to the air data computer (ADC) to determine the actual state and share the information with other on-board systems. Some aircraft are equipped with other sensors which can measure humidity

(mixing ratio) and/or sensors to detect the presence of ice on the flying surfaces can be measured. An inertial reference platform is part of the equipment for normal, longitudinal and lateral acceleration and rotations.

The Flight Management System (FMS) uses the information from the sensors for flight safety and cockpit information systems. Modern aircraft are equipped with a positioning system which exploits the information from the Global Navigation Satellite System (GNSS). The on-board computer combines parameters to determine the position of the aircraft and comple-

ment the other sensors. For example, the Mach number is computed using static and total pressure measurements obtained by the pitot-tube. This Mach number is used in the derivation of the static air temperature derived from total air temperature. The true airspeed is in turn calculated using the Mach number and the total air temperature. Wind vector information is computed using the air vector (true airspeed and heading) and the ground vector (ground speed and track angle).

### 2.1   Mach number measurement

One of the basic instruments on an aircraft is the pitot-tube which is used to determine the Mach number by measuring the static pressure $p_s$ and impact pressure $q_i$ (which is the difference between static pressure $p_s$ and total air pressure $p_t$, $q_i = p_t - p_s$).





The Mach number is calculated as follows,

$$M = \sqrt{\frac{2}{\gamma-1}\left(\left(\frac{q_i}{p_s}+1\right)^{\frac{1}{\gamma}(\gamma-1)}-1\right)}, \tag{1}$$

where $\gamma$ is the ratio of specific heats of dry air ($c_p$ and $c_v$).

The pressure measurements are deteriorated by the so-called pressure defect (Rodi and Leon, 2012), which is caused by flow disturbances around the sensor, and depends on the angle of attack and the airspeed. Both total and static pressure suffer from this. The impact pressure is more accurate because it is the difference of the two.

    The Mach number is the quotient of the true airspeed $V_a$ of the aircraft and the speed of sound $c$. The latter depends on the ambient static air temperature (neglecting the small contribution of humidity) by

$$c = \sqrt{\gamma R T} = \sqrt{\gamma \frac{p}{\rho}}, \tag{2}$$

with $R = 287.05$ J/K/kg the gas constant of dry air, and to estimate the true airspeed, temperature information is thus needed. A measurement related to true airspeed is the indicated airspeed,

$$V_I = \sqrt{\frac{p_0}{\rho_0}\frac{2\gamma}{\gamma-1}\left(\left(\frac{q_i}{p_0}+1\right)^{\frac{1}{\gamma}(\gamma-1)}-1\right)} \tag{3}$$

which is the true airspeed at sea level ($p_0 = 1013.25$ hPa and $\rho_0 = 1.225$ kg/m$^3$).

## 2.2   Aircraft temperature measurement

A thermometer probe measures the (stagnation) temperature $T_i$. This temperature is in general not equal to the static air temperature $T_a$ due the stagnation of the air and viscosity effects. The static air temperature, is related to the measured temperature by (following Painting (2003))

$$T_a = T_i\left(1 + \lambda\frac{\gamma-1}{2}M^2\right)^{-1}, \tag{4}$$

where $M$ is the Mach number, and $\lambda$ is the probe recovery factor, which includes the effect of viscosity, and the effect of incomplete stagnation of air at the sensor. For the most common probe in service on commercial aircraft, $\lambda = 0.97$, and given $\gamma = 1.4$, the static air temperature becomes

$$T_a = T_i/(1 + 0.194 M^2). \tag{5}$$

## 2.3   AMDAR observations

Using the AMDAR system, a selection of the information which is available in the on-board computer can be transmitted to a ground station. The AMDAR software installed on the on-board computer collects and transmits the information through the ACARS system. In this way, wind and temperature observations can be received almost real-time, even from remote areas.



**Table 1.** Parameter resolution of time and position of AMDAR and Mode-S EHS.

|  | AMDAR | Mode-S EHS |
|---|---|---|
| temperature | 0.1 K | - |
| time | 60 s | 1s - 1ms |
| latitude | 0.001 deg | 0.0001 deg |
| longitude | 0.001 deg | 0.0001 deg |
| Mach number | - | 0.004 |
| true airspeed | - | 2 kt ≈ 1 m/s |
| indicated airspeed | - | 1 kt ≈ 0.5 m/s |
| barometric altitude | 3.2 ft ≈ 0.97m | 25 ft≈ 7.6 m |
| barometric altitude rate | - | 32 ft/min ≈ 0.16 m/s |

## 2.4 Mode-S EHS observations

Some information, available in the on-board computer, can be extracted from down-linked air traffic control (ATC) information
using the secondary surveillance radar (SSR) technique Mode-S EHS. In the European designated EHS airspace, all fixed wing
aircraft, having a maximum take-off mass greater than 5,700 kg or a maximum cruising true airspeed in excess of 125 m/s
(approx. 250 knots), must be Mode-S EHS compliant and should respond to the radar request. The set of parameters that
can be down-linked consists of Mach number, air speed, indicated airspeed, magnetic heading and roll angle. In this study,
the Mach number will be used to investigate the AMDAR temperature measurement accuracy. See de Haan (2011) for more
details.

## 2.5 Numerical weather prediction model data

The used numerical weather prediction (NWP) model data is from the operational non-hydrostatic model HIRLAM (Undén
et al., 2002) which is run 8 times per day with a 3DVAR assimilation cycle. AMDAR data is used in the assimilation but
to avoid problems in the comparison a forecast with a lead time of at least 3 hours is used. The NWP model equivalent of
the AMDAR temperature observation is determined by bi-linear interpolation in the horizontal and linear interpolation in the
logarithm of the pressure. A linear interpolation in time is performed between hourly space interpolated positions.

## 2.6 Parameter resolution and collocation method

In Table 1 the reported resolution of time and position of AMDAR and Mode-S EHS are given. Apart from altitude, Mode-S
EHS location parameters are reported at a higher resolution. Especially the time resolution of AMDAR is low with respect to
the altitude resolution. Note that the barometric altitude rate, the vertical speed of the aircraft, is not available in the AMDAR
message.





The AMDAR and Mode-S EHS data sets are collocated as follows. The WMO AMDAR aircraft identifier, provided in the reports is linked with a ICAO 24-bit identifier using a lookup-table. Next, within 300 s after an AMDAR observation, two Mode-S EHS observations are identified for which the pressure altitudes are closest, with one smaller and one larger than the

AMDAR pressure altitude.

The lookup-table has been provided by the Eumetnet AMDAR Programme Management for a number of aircraft; for the aircraft for which no official link between tail-number (or ICAO 24-bit identifier) is available a collocation query has been applied. The result of this query is validated against the provided lookup-table.

## 3  Temperature error sources

In this manuscript it is assumed that the temperature error can be separated into a flight phase dependent, a Mach-number and (static) pressure related part. The reference temperature used stems from NWP and has its own characteristics but is assumed to be flight phase and Mach-number independent. In this section the methodology to determine the temperature bias characteristics from the observations is presented.

### 3.1  Flight phase dependent bias

It is observed that some aircraft exhibit a different bias when descending and ascending. Since generally an atmospheric profile has a temperature lapse rate of $\Gamma$=-6.5 K/km =-0.0019812 K/ft this bias could be caused by time mis-synchronization between height message and the temperature message.

When the observation time of the temperature and the height differ, a bias will be introduced with opposite sign for descending and ascending flight paths. Suppose the time difference is $\tau$, that is when temperature $T$ is observed at $t_T$ and the height at

$t_h$, with $t_T = t_h - \tau$, then

$$h(t_T) = h(t_h - \tau) \approx h(t_h) - \tau v, \tag{6}$$

where $v$ is the aircraft vertical speed. The height difference between reported height and height of the temperature is $v\tau$ and thus a bias of $\Gamma v\tau [K]$ will be present, that is

$$T(t_h) = T(t_T + \tau) \approx T(t_T) + \tau \frac{dT}{dt} = T(t_T) + \tau v \frac{dT}{dh}. \tag{7}$$

Inversely, when we know the vertical lapse rate and the temperature bias we can estimate the time difference. The time difference is assumed to be the same for descending and ascending flight paths and thus the bias due to the time difference is of opposite sign and different magnitude due to the difference in vertical velocity when descending or ascending. This implies that when the sum of the time difference biases is not equal to zero the temperature is really biased assuming that the bias not related to the time difference, is independent of the flight phase. This is most likely the case when the observation is compared

to model temperatures, because of representativeness, model orography and/or model parametrization.

Next, we estimate the time difference. Let $\tau$ be the time difference, $v_d$ and $v_a$ be the descending vertical velocity, respectively the ascending vertical velocity. The observed descending and ascending temperatures $T_d$ and $T_a$ will differ from the measured





temperature $T$ (without the time difference) as follows

$$T_d = T + \tau v_d \frac{dT}{dh} \tag{8}$$

$$T_a = T + \tau v_a \frac{dT}{dh}. \tag{9}$$

Suppose we have another observation of temperature $T^{ref}$ (e.g. from a numerical weather prediction model), which deviates from the AMDAR temperature $T$ by

$$T^{ref} = T + \beta + \epsilon, \tag{10}$$

where $\beta$ is the bias between AMDAR and the reference and $\epsilon$ the part of the temperature difference that cannot be described by $\beta$ and which has a zero mean value. Let $\Delta T$ be the difference between observation and the reference, then

$$\Delta T_d = \tau v_d \frac{dT}{dh} - \beta - \epsilon \tag{11}$$

$$\Delta T_a = \tau v_a \frac{dT}{dh} - \beta - \epsilon, \tag{12}$$

The time difference $\tau$ can be found as using many of collocated observations (in which case $\bar{\epsilon}=0$)

$$\tau = \frac{\overline{\Delta T}_d - \overline{\Delta T}_a}{\overline{v}_d \Gamma - \overline{v}_a \Gamma}, \tag{13}$$

where $\overline{x}$ denotes the mean value of parameter $x$.

## 3.2 Static pressure correction

The impact pressure $q_i$ is measured accurately, but the static pressure needs correction for angle of attack, airspeed and possible other state variables (Rodi and Leon, 2012). Let $\tilde{p}_s$ be the uncorrected static pressure and the function $f$ describes the correction of the static pressure, that is

$$p_s = f(\tilde{p}_s, V_a, ...). \tag{15}$$

The corrected $p_s$ is used to determine a corrected Mach number and true airspeed.

Because we observe that the temperature observation exhibits bias, we assume that this parameter is not recalculated using a corrected Mach number, that is

$$\tilde{T}_a = T_i \left(1 + \lambda \left(\left(\frac{q_i}{\tilde{p}_s} + 1\right)^{\frac{\gamma-1}{\gamma}} - 1\right)\right)^{-1}, \tag{14}$$

where $\tilde{T}_a$ is (uncorrected) biased temperature.

Suppose now that we can find an estimate of the inversion of correction function $f$, then together with an estimate the dynamic pressure $q_i$ deduced from indicated airspeed, we can estimate the impact temperature $T_i$ by

$$T_i = \tilde{T}_a \left(1 + \lambda \left(\left(\frac{q_i}{f^{-1}(p_s, V_a, ...)} + 1\right)^{\frac{\gamma-1}{\gamma}} - 1\right)\right), \tag{15}$$





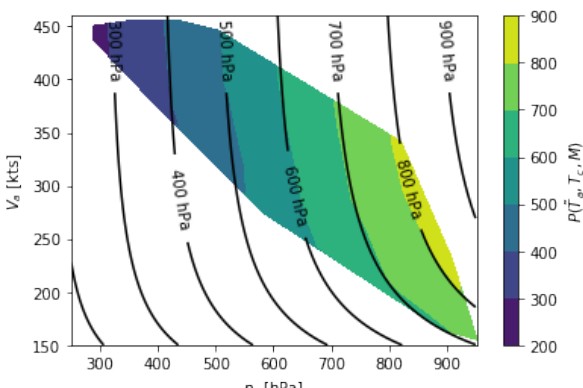

**Figure 1.** Example of $P(\tilde{T}_a, T_c, M)$ for aircraft EU0884 (filled contour) and the approximation by $f^{-1}$.

and with the (corrected) Mach number, we can estimate the static air temperature as

$$
\quad T_a = T_i \left(1 + \lambda \frac{\gamma-1}{2} M^2\right)^{-1} = \tilde{T}_a \left(1 + \lambda \left(\left(\frac{q_i}{f^{-1}(p_s, V_a, ...)} + 1\right)^{\frac{\gamma-1}{\gamma}} - 1\right)\right) \left(1 + \lambda \frac{\gamma-1}{2} M^2\right)^{-1}, \tag{16}
$$

Thus, when we have an (estimate) of the mapping $f^{-1}$ we can correct the temperature measurement.

Suppose now that we have a set of collocated temperatures over a long period and full pressure range, then we might be able to find the inversion of the function $f$. Let $T_c$ be the temperature used for calibration. Then $f$ should obey

$$
f^{-1}(p_s, V_a, ...) \approx P(\tilde{T}_a, T_c, M), \tag{17}
$$

where

$$
P(\tilde{T}_a, T_c, M) = q_i \left(\left(\left(\frac{\tilde{T}_a}{T_c}\left(1 + \lambda \frac{\gamma-1}{2} M^2\right) - 1\right)\lambda^{-1} + 1\right)^{\frac{\gamma}{\gamma-1}} - 1\right)^{-1} \tag{18}
$$

Figure 1 shows the value of $f^{-1}$ as a function of $p_s$ and $V_a$ for a selected aircraft (filled contours) using NWP data over an 18 months period. As it turns out, the most dominant terms are related to pressure and true airspeed. The fit was constructed by binning both $p_s$ and $V_a$ in 10 separate bins, and use the median value of $f^{-1}$ in the least squares fit. The function $f^{-1}$ is

approximated by

$$
f^{-1}(p_s, V_a) = a + b p_s + c \frac{p_s}{V_a^2} \tag{19}
$$

To avoid extreme values of $b$, we constrained $b$ to the interval $(0.8, 1.2)$. We observe that the chosen representation of $f^{-1}$ (contour lines) fits the data (filled contour).





## 4 Data preparation

### 4.1 Estimation of correction coefficients

We collocated AMDAR temperature observatories with NWP for the period from 2017/1/2 to 2018/7/31. We used a forecast lead time of at least 3 hours to avoid correlation due to assimilation of AMDAR. Next, the AMDAR observations are collocated with Mode-S EHS observations, where the height of observations is primarily used for collocating, because the AMDAR time resolution is minutes which is too coarse; the Mode-S EHS observations are linearly interpolated with respect to the AMDAR reported height. The observation time difference is at most 120 seconds. The fit is performed on binned data where the median

values in bins are used. In this way we avoid over-fitting for pressure-airspeed values that occur more frequent than others. Furthermore, a minimum of 10 data points per bin was required.

### 4.2 Collocation with Radiosonde observations

To show that the two correction methods have a positive effect on the measurement accuracy and bias of AMDAR temperatures

we compared the uncorrected and corrected values with independent observations. Over a period from 2018-08-10 to 2019-12-17 AMDAR observations are collocated with radiosonde observations. This period has no overlap with the period used to determine the correction coefficients. Radiosonde are generally considered as a profile, at one location and a single timestamp, but they are not. The time the balloon needs to reach 500hPa is around 20 minutes, while the wind carries the platform over a distance of sometimes more than 100 km.

Radiosondes are generally launched at the main hours (00, 06, 12, 18 UTC), as required by WMO, with the majority of launches around 00 and 12 UTC (these timestamps represent the observation at a level of 500 hPa at the whole hour)

The data set used was received over the GTS, and contains the operational available observations with a high resolution in reporting time (sometimes every second). All observations had a single timestamp which (should) represent the moment the balloon reaches 500hPa. The observation time was altered to take into account that the balloon rises with a vertical speed of

approximately 5 m/s. We did not include horizontal drift of the balloon.

An AMDAR observation is collocated with a radiosonde observation when the distance is smaller than 50 km, the time difference is smaller than 30 minutes and the height difference is less than 15 m. For each AMDAR observation, a nearby radiosonde observation, if exists, was found. This implies that a radiosonde observation could have multiple matching AMDAR observations. This is reasonable for this study since we are interested in the quality of (corrected) AMDAR observations.

## 200  5 Results

In this section we discuss the results of correcting AMDAR temperature observations by reconstruction of the uncorrected static pressure. The corrections are derived using NWP data over a period of 17 months (January 2017 to July 2018). The corrections are applied to AMDAR observations from the period September 2018 to December 2019. The (un)corrected temperatures are compared to radiosonde observations.



**Table 2.** Statistics of uncorrected AMDAR temperature observations (raw) minus radiosonde temperature observations, for the period 2018/9/17 to 2019/12/17. Statistics of corrected AMDAR: only flight phase correction ($\tau$), only pressure correction ($p_s$), and both corrections applied ($\tau\,p_s$). The corrections are estimated using NWP and Mode-S EHS from the period 2017/1/2 to 2018/7/31. The number of collocated data points is 14716.

| correction method | mean | standard deviation |
|---|---|---|
| raw | 0.389 | 1.007 |
| $\tau$ | 0.343 | 1.007 |
| $p_s$ | 0.049 | 0.923 |
| $\tau, p_s$ | 0.003 | 0.921 |

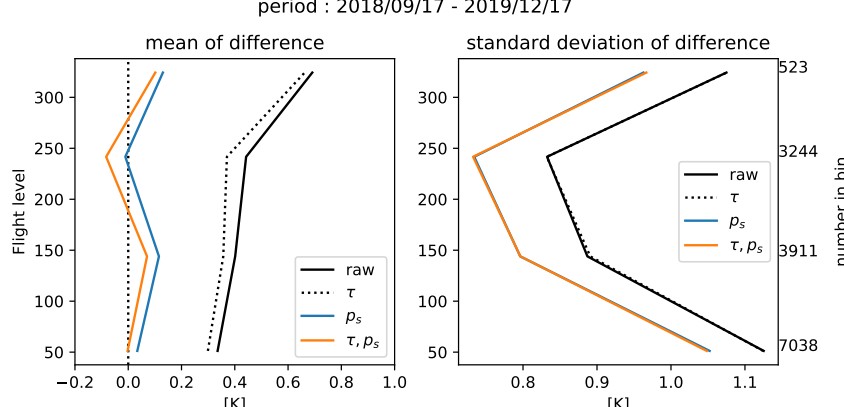

**Figure 2.** Statistics of AMDAR temperature minus radiosonde temperature.

Both time periods and the source of information do not overlap and are independent implying a safe and sound comparison. Table 2 shows the result of the comparison. Clearly, the warm bias is diminished by applying both corrections. But not only the bias has improved; the standard deviation improves by almost 10%. The magnitude of the standard deviation is higher than previously reported (0.6 K) because the collocation is less tight. Especially large difference in time and distance increase the standard deviation.

Figure 2 shows the mean difference (left panel) and standard deviation of the difference (right panel) with respect to flight level. Over the whole atmospheric profile, the bias and standard deviation improve significantly when the corrections are applied. Note the numbers on right panel denote the number of data points in each vertical bin.

Figure 3 shows 4 panels with profile statistics for the main synoptic hours. The most left panel shows that at 00 UTC, after correction, the bias has an almost constant but small positive value. The most right panel (18 UTC) has the most positive bias





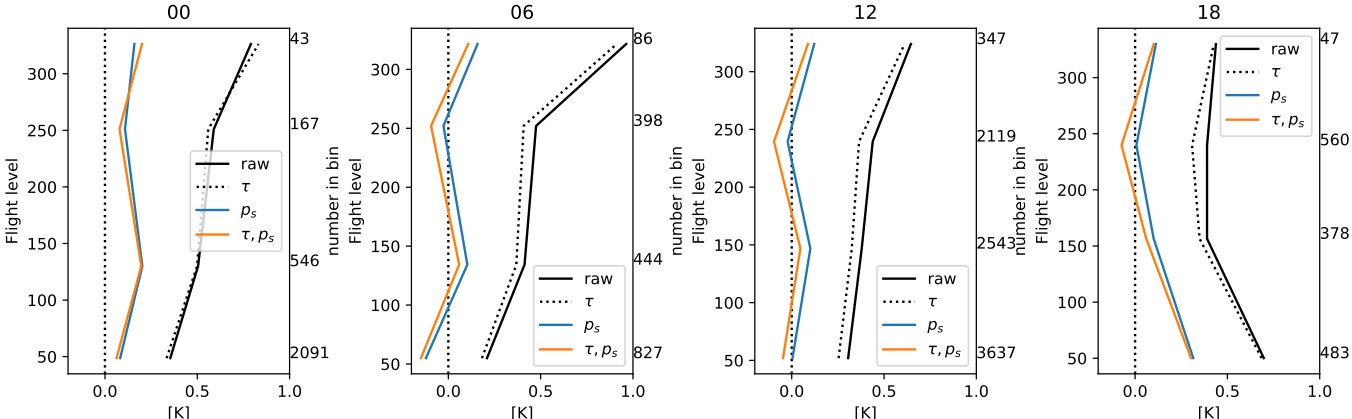

**Figure 3.** Mean of AMDAR temperature minus radiosonde temperature, subdivided into four synoptic hours.

below flight level 150. The bias is around zero for 12 UTC observations The reason for the difference in bias with the time of day is not understood. Assuming that the AMDAR bias is constant we observe that the radiosonde bias, changes over the day from overestimation at 06 UTC to neutral at 12 UTC and underestimation at 18 UTC to slightly underestimation at 00 UTC.

Figure 4 shows the mean (left panel) and standard deviation (right panel) of the difference between AMDAR and radiosonde temperature observations grouped by aircraft type. A minimum threshold of 10 observations per aircraft is required. For all but three aircraft types, the bias is reduced. For (a single) A30B we observe that the bias changes sign, for the 7 A388 aircraft we observe no change, and for the 2 B736 a slight increase is observed. The reduction in bias for aircraft with large biases is very large. With respect to the standard deviation all observed standard deviations are equal or smaller with corrections applied.

## 6 Conclusions

In this manuscript we demonstrated that the AMDAR warm bias can be characterized by two methods of corrections: the first is a timing related correction, while the second is an accuracy related correction. Both corrections can be found using an external source of temperature information; together with Mode-S EHS down linked parameters, such as true airspeed and Mach number.

In this manuscript we used NWP data to characterize the corrections. Also, the corrected AMDAR temperatures were compared to radiosonde observations but for a different period, so that this comparison was completely independent. As a consequence the resulting bias was diminished by the correction, while the standard deviation reduced by almost 10 %.

Both corrections are currently assumed to be constant and static in time. To assure model or source independence, a different dataset in time is required to construct the correction parameters ($\tau$, $a$, $b$ and $c$). Further research, including a longer time period, is required to verify that this constant assumption is valid since one can expect that aircraft maintenance can affect the



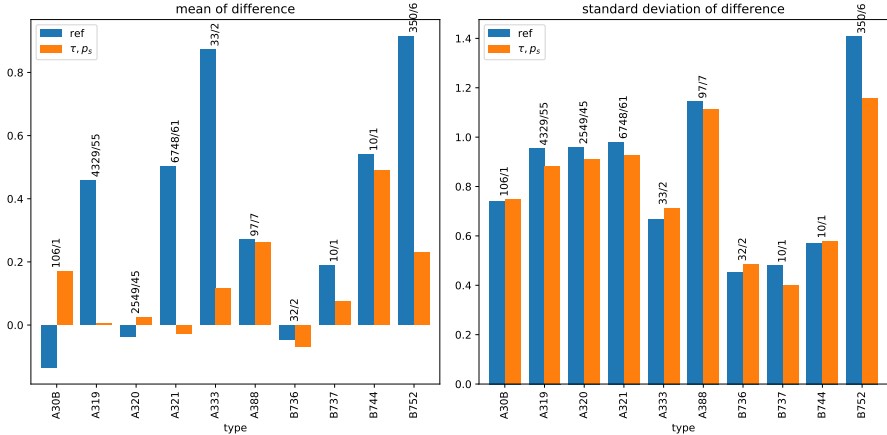

**Figure 4.** Bias and standard deviation of AMDAR minus radiosonde temperatures, grouped by aircraft type. The two numbers for each group denote the total number of observations and the number of unique aircraft, respectively.

time synchronization or static pressure corrections. This could also explain the results for the two aircraft showing increased
235   bias in our results.

The Mode-S EHS information can be applied to correct the AMDAR temperature bias, for those air spaces where Mode-S EHS information is available.

*Author contributions.* The main research was carried out by dr. de Haan; Dr. de Jong contribution was mainly related to Mode-S, while dr. van der Meulen's contribution was on AMDAR.

240   *Competing interests.* There are no competing interests present.

*Acknowledgements.* The authors would like to thank the Eumetnet AMDAR Programme Management for providing the aircraft identifier lookup-table. The Mode-S EHS data used in this study has kindly been provided by EUROCONTROL, Maastricht Upper Area Control.



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
