# Peer review of "Characterizing and correcting the warm bias observed in AMDAR temperature observations"

_Atmospheric Measurement Techniques, 2020_

## Referee Comment (RC1)

**General review of the manuscript AMT-2020-519**

Title: "Characterizing and correcting the warm bias observed in AMDAR temperature observations"

Authors: Siebren de Haan, Paul M. A. de Jong, and Jitze van der Meulen.

Manuscript Number: AMD-2020-519

Reviewer #2: Mikhail A. Strunin

Recommendation: Manuscript can be published in AMT after appropriate revision

**General comments**

The presented manuscript is related to an important and relevant topic of obtaining correct data on vertical distributions of air temperature using the AMDAR systems installed on commercial airplanes. As the authors note numerous investigations have shown that there are systematic discrepancies between the air temperature measured by the AMDAR system during airplane descending and climbing with the data of radio-sounding data and weather prediction models. The authors explain these discrepancies through the time lag between the temperature readings of the AMDAR system during aircraft climbing or descending and the radio-sounding observations in the temperature-stratified atmosphere. To eliminate this systematic error, the authors applied the original method for correcting AMDAR temperature measurements, based on a statistical analysis of the differences between the AMDAR system readings and radio-sounding data.

Undoubtedly, the statistical approach has the right to use, however, a number of uncertainties in the manuscript require several remarks regarding the physical base for the reasons of the delay in the AMDAR system. This is important, since the authors do not consider the nature of the arising temperature lags, limiting their research by a purely statistical approach.

It seems that the reason for the time lags is the inertia of the temperature and pressure sensors used in the AMDAR system. Typically commercial aircraft are equipped with Rosemount 102 temperature sensors of various modifications, whose thermal inertia is about 1 to 2 seconds, depending on the flight conditions. Measurements of pressure, both total and static, also have some inertial properties. In this case, the main reason for the delay in the pressure readings relative to the true altitude is the time lag in the air pipes connecting the pressure receivers (Pitot tube) with the pressure transducer installed inside the aircraft fuselage. Obviously, the value of time lag will also depend on the type of aircraft, because the configuration of the location of the sensors could be different on different types of aircraft.

The authors also do not consider possible distortions of the air flow nearby the aircraft fuselage skin during aircraft maneuvers, when the pressure and temperature receivers, which are mounted on the skin, could fall into the zone of the so-called "aerodynamic shadow". This could significantly distort the readings of the sensors. The authors are limited themselves to introducing corrections to pressure readings using some universal function. However, such a method can give a completely satisfactory result, but a fine analysis of errors, perhaps, is not beyond the scope of this manuscript.

Generally, the manuscript is of undoubted interest, since the suggested methods could improve the quality of assimilated AMDAR data for meteorological models and prognostic programs. Below the number of remarks are presented, some of which are of principled importance for the subject of the manuscript.

**Specific comments**

**Major specific remarks**

Line 120 and bellow. The authors describe the method for determining the time delay of the air temperature recording in the AMDAR system. The very value of the correction to the temperature cannot be large. With a typical time constant for Rosemount 102 sensors of 1 s and an airplane ascent or descent rate of 10 ms-1 (this value even exceeds the standard rate of climb of civil aircraft of 5 ms-1), as well as with standard temperature stratification in the atmosphere in 0.0065 K/m, the correction will be no more than 0.065 K. It is the values of the correction that were obtained by the authors (see Figures 2 and 3).

However, the question arises, what will be the magnitude of the correction and its sign for unstable temperature stratification in the atmosphere? Super-adiabatic temperature gradients are often observed in the lower part atmosphere, in the surface and boundary layers.

From the description provided by the authors, it is also unclear what altitude is used to correct the air temperature, barometric or geometric, based on the global positioning system. This is important in this case, since the registration of the barometric altitude also occurs with a time delay relative to the true altitude due to the pressure inertia of the aircraft air pipe.

Line 150 and bellow. The procedure presented by the authors for the AMDAR pressure processing has the goal for correcting its value, taking into account the distortion of the air flow depending on the speed and altitude of flight, which is quite fair. However, the proposed method does not allow estimating the value of the pressure readings lag in the AMDAR system, which was declared by the authors. Authors need to clarify these aspects of the study.

Line 200 and bellow. Figures 2 and 3 show the vertical profiles of the discrepancy between the AMDAR air temperature readings and the radio-sounding data. Obviously, the authors

succeeded in significantly diminishing of the discrepancies between different observation methods. At the same time, the manuscript implicitly declares that the proposed correction methods make it possible to exclude systematic errors of determining the air temperature in the AMDAR system. However, as it follows from the figures that the observed peaks on the profiles represent a systematic error. If the residual error were random, then the profiles of the discrepancies would have chaotic peaks. The systematic character of the discrepancies is also confirmed by the profile of standard deviations in Figure 2. The maxima on these profiles correspond to the heights where peaks are observed on the profiles of discrepancies. It would be desirable to clarify the nature of the residual systematic discrepancies.

**There are also a few minor specific remarks:**

line 165 and bellow. Figure 1 provides a diagram for evaluating corrections to pressure measured by the AMDAR system. The aircraft number is given, but its type is not indicated. The question arises as to how applicable this diagram is to other types of aircraft.

line 165 and bellow. Approximation formula (19) is presented without justification; it is necessary to clarify at what values of the parameters it is valid and what is the possible error.

line 185 and bellow. The authors do not indicate the region (or rgions) where the comparisons of the aircraft and radio-sounding data were fulfilled, as well as the weather conditions during comparisons. Has the temperature stratification always been close up to the standard?

line 200 and bellow. Путаница на рисунках 2 и 3, поправка на запаздывание по температуре и вертикальная нулевая ось на рисунках обозначены практически одинаковыми линиями.

line 200 and bellow. There is confusion in Figures 2 and 3 and the profiles of temperature lag correction and the vertical zero axes in the figures are indicated by almost the same lines.

**Technical corrections**

There are many technical confusing of the terms, concerning conventional aerodynamic and aerospace techniques, pictures and tables do not have all necessary titles.

line 55 (and hereinafter): it is need to use the conventional aerodynamic term "dynamic pressure" instead of "impact pressure".

line 65: There is no need for formula (2) to determine the speed of sound, the Mach number is well-known parameter.

line 65: Formula (3) for the indicated air speed does not make sense for this manuscript, since it is necessary to use the true aircraft airspeed taking into account the current air temperature.

line 70 (and hereinafter): it is need to use the conventional aerodynamic term "total air temperature" instead of "stagnation temperature".

line 80, Table 1: not "Parameter resolution of time and position", but "Time and position resolution". What is time resolution "1 s - 1 ms"?

line 115: (and hereinafter): it is need to use the conventional meteorological term "vertical temperature gradient" or "temperature stratification" instead of "lapse rate".

line 155, table 2: It is need to indicate dimensions of mean and standard deviation values in the title of the table.

line 155, figure 1: What is the type of aircraft EU0884? This is important because the correction diagram depends on type of aircraft. Titles of axes in figure should be larger.

line 200, figure 2: Labels of the axis of height is not indicated. Title of temperature axis is not shown. What is the "number in bin"? Titles of axes in figure should be larger. It also seems that lines of "tay" and vertical zero axis are confused.

line 210, figure 3: Labels of the axis of height is not indicated. Title of temperature axis is not shown. What is the "number in bin"? It is needed to indicate "synoptic hours" in top of the axis. Titles of axes in figure should be larger. It also seems that lines of "tay" and vertical zero axis are confused.

Authors should improve English language of the manuscript. I would advise authors to refer to native speakers English speakers to correct the text.

**Conclusion:**

Manuscript "Characterizing and correcting the warm bias observed in AMDAR temperature observations" presented by Siebren de Haan, Paul M. A. de Jong, and Jitze van der Meulen can be published in AMT after appropriate revision.

---

## Referee Comment (RC2)

Referee Report for
**Characterizing and correcting the warm bias observed in AMDAR temperature observations**

The correction approach for AMDAR observations is well documented. Furthermore, their test of the correction appears to be carefully conducted and indicates a measurable improvement when compared to independent observations. I therefore recommend publication of the manuscript in Atmospheric Measurement Techniques. However I did find some minor corrections and comments that the authors may wish to address:

1. Some nomenclature is not fully defined in the text. The ones I noted were:

   - In equation 2, $T$, $\rho$ and $P$ are not defined. In addition, it should be clarified whether these are total or static temperatures/pressures. Also, since the static temperature and pressure is defined as $T_a$ and $p_s$, it may be more clear to use these terms in this equation.
   - Similarly, in the development of the correction, $T_a$ is is used for ascending temperature, whereas it has previously been defined for static air Temperature. In the same section $T$ is introduced as simply temperature, without clearly indicating whether the authors are referring to total or static temperature.
   - $V_a$ is not defined, and appears in Section 3.2.
   - The acronym ECMWF is not defined.
   - The acronym ACARS is not defined.
   - The acronlym WCO is not defined.

2. In Figures 2 and 3 the same dotted line is used for the $\tau$ correction and the zero line. The line format for one of these should be changed to avoid confusion.

3. The two phenomena that are being corrected are not the only ones which may provide warm bias, and the authors may wish to provide potential sources of bias which may affect the warm bias, but are not as easy to correct for (For example, calibration bias/sensor drift and inaccuracy in $\lambda$).

4. Given the results presented in Figure 4, is it possible to correct for aircraft dependent bias through $\lambda$?

---

## Author Comment (AC1)

We kindly thank the reviewer for his comments and questions.
Below you will find our response to these comments and questions together with questions and comments themselves.
Kind regards,
Siebren de Haan , et al.

5  *Line 120 and bellow. The authors describe the method for determining the time delay of the air temperature recording in the AMDAR system. The very value of the correction to the temperature cannot be large. With a typical time constant for Rosemount 102 sensors of 1 s and an airplane ascent or descent rate of 10 ms-1 (this value even exceeds the standard rate of climb of civil aircraft of 5 ms-1), as well as with standard temperature stratification in the atmosphere in 0.0065 K/m, the correction will be no more than 0.065 K. It is the values of the correction that were obtained by the authors (see Figures 2 and 3).*

10  » This is indeed the magnitude of the first correction.

*However, the question arises, what will be the magnitude of the correction and its sign for unstable temperature stratification in the atmosphere? Super-adiabatic temperature gradients are often observed in the lower part atmosphere, in the surface and boundary layers.*

» We do not correct the temperature measurement, but the reference height, assuming the delay is independent of the weather

*From the description provided by the authors, it is also unclear what altitude is used to correct the air temperature, barometric or*

15  *geometric, based on the global positioning system. This is important in this case, since the registration of the barometric altitude also occurs with a time delay relative to the true altitude due to the pressure inertia of the aircraft air pipe.*

» We added the following :Finally, we correct the reference height of the temperature measurement using the vertical velocity of the aircraft."

*Line 150 and bellow. The procedure presented by the authors for the AMDAR pressure processing has the goal for correcting its value,*

20  *taking into account the distortion of the air flow depending on the speed and altitude of flight, which is quite fair. However, the proposed method does not allow estimating the value of the pressure readings lag in the AMDAR system, which was declared by the authors. Authors need to clarify these aspects of the study.*

» The pressure lag, when present, is included in the correction.

*Line 200 and bellow. Figures 2 and 3 show the vertical profiles of the discrepancy between the AMDAR air temperature readings and*

25  *the radio-sounding data. Obviously, the authors succeeded in significantly diminishing of the discrepancies between different observation methods. At the same time, the manuscript implicitly declares that the proposed correction methods make it possible to exclude systematic errors of determining the air temperature in the AMDAR system. However, as it follows from the figures that the observed peaks on the profiles represent a systematic error. If the residual error were random, then the profiles of the discrepancies would have chaotic peaks. The systematic character of the discrepancies is also confirmed by the profile of standard deviations in Figure 2. The maxima on these profiles*

30  *correspond to the heights where peaks are observed on the profiles of discrepancies. It would be desirable to clarify the nature of the residual systematic discrepancies.*

» the increase of standard deviation near the surface is related to natural variability of temperature and the fact that both measurements are not completely Colocated. The larger standard deviation near the top could be related to general measurement inaccuracies of the aircraft temperature measurement

35  *line 165 and bellow. Figure 1 provides a diagram for evaluating corrections to pressure measured by the AMDAR system. The aircraft number is given, but its type is not indicated. The question arises as to how applicable this diagram is to other types of aircraft.*

» This diagram is just an example; diagrams of other aircraft showed a similar picture.

*line 165 and bellow. Approximation formula (19) is presented without justification; it is necessary to clarify at what values of the parameters it is valid and what is the possible error.*

40  » the justification is stated in the lines preceeding this equation.

*line 185 and bellow. The authors do not indicate the region (or rgions) where the comparisons of the aircraft and radio-sounding data were fulfilled, as well as the weather conditions during comparisons. Has the temperature stratification always been close up to the standard?*

» we added information on the geographical coverage to the text

*line 200 and bellow. There is confusion in Figures 2 and 3 and the profiles of temperature lag correction and the vertical zero axes in the*

45  *figures are indicated by almost the same lines.*

» changed the linestyle of the axes

*line 55 (and hereinafter): it is need to use the conventional aerodynamic term "dynamic pressure" instead of "impact pressure".*

» wording is changed

*line 65: There is no need for formula (2) to determine the speed of sound, the Mach number is well-known parameter.*

» We added some words on the measurement of temperature using Bernoulli's equation. In this derivation the speed of sound is needed

*line 65: Formula (3) for the indicated air speed does not make sense for this manuscript, since it is necessary to use the true aircraft airspeed taking into account the current air temperature.*

» adjusted accordingly

*line 70 (and hereinafter): it is need to use the conventional aerodynamic term "total air temperature" instead of "stagnation temperature".*

» adjusted accordingly

*line 80, Table 1: not "Parameter resolution of time and position", but "Time and position resolution". What is time resolution "1 s - 1 ms"?*

» adjusted accordingly

*line 115: (and hereinafter): it is need to use the conventional meteorological term "vertical temperature gradient" or "temperature stratification" instead of "lapse rate".*

» adjusted accordingly

*line 155, table 2: It is need to indicate dimensions of mean and standard deviation values in the title of the table.*

» adjusted accordingly

*line 155, figure 1: What is the type of aircraft EU0884? This is important because the correction diagram depends on type of aircraft. Titles of axes in figure should be larger.*

» adjusted accordingly

*line 200, figure 2: Labels of the axis of height is not indicated. Title of temperature axis is not shown. What is the "number in bin"? Titles of axes in figure should be larger. It also seems that lines of "tay" and vertical zero axis are confused.*

» adjusted accordingly

*line 210, figure 3: Labels of the axis of height is not indicated. Title of temperature axis is not shown. What is the "number in bin"? It is needed to indicate "synoptic hours" in top of the axis. Titles of axes in figure should be larger. It also seems that lines of "tay" and vertical zero axis are confused.*

» adjusted accordingly

---

## Author Comment (AC3)

We kindly thank the reviewer for his comments and questions.
Below you will find our response to these comments and questions together with questions and comments themselves.
Kind regards,
Siebren de Haan , et al.

5    *line 17 'Upper air observations from aircraft are an important source of information for numerical weather prediction (NWP).' Possibly reference Ingleby et al (2021).*
     » Done: added reference
     *line 21,22 'ECMWF has introduced an aircraft and flight phase dependent temperature correction (Cardinali et al., 2004).' The ECMWF bias correction of aircraft temperatures was announced in a short newsletter item by Isaksen et al (2012, see below), an update is given in*
10  *Ingleby et al (2019). Cardinali et al (2004) did -not- discuss aircraft temperature biases. I think the work at NCEP by Zhu et al (2015) should also be mentioned.*
     » Done: added references
     *line 29 'The formal difference is slightly smaller than 0.4K (Painting, 2003).' I'm not sure what this means - clarify or delete. It might be worth mentioning the EUFAR workshop on aircraft temperature measurements (Nov 2020): https://www.eufar.net/shared-subjects/s/3fa8510de42844d6b3*
15  *(I particularly remember the presentation by Bob Sable on TAT sensors - it seems the industry preoccupation is with avoiding icing of the sensors in extreme conditions and accuracy was a secondary consideration.)*
     » Deleted
     *line 40 'Aircraft sensors' I recommend that the book by Wendisch and Brenguier (2012) is referenced in this section.*
     » we added text to clarify
20  *line 70ff 'Aircraft temperature measurement' This is key and should probably be expanded slightly - to mention the convertion of kinetic energy to temperature (mainly by adiabatic compression within the TAT probe). Perhaps mention typical differences between Ta and Ti. Section 2.5 of Wendisch and Brenguier is useful - it derives equations like (4). WMO seems to be encouraging use of WMO No. 8 'Guide to Meteorological Instruments and Methods of Observation' rather than Painting (2003).*
     » Done: used the proposed reference
25  *line 91 '2.5 Numerical weather prediction model data' Either here or earlier the geographical domain being used should be mentioned.*
     » Done: added a description of the region
     *line 115,116 'Since generally an atmospheric profile has a temperature lapse rate of -6.5 K/km' 'Since average tropospheric profiles have ...' would be more accurate.*
     » Done
30  *line 123,124 'the temperature is really biased assuming that the bias not related to the time difference, is independent of the flight phase' perhaps 'there is an additional bias term, which may be independent of the flight phase'*
     » Done
     *line 145 What is the typical time difference tau? How much does it vary? Is it linked to aircraft type and/or airline?*
     »
35  *line 147 'possible' - 'possibly'*
     » Done
     *line 161 'Thus, when we have an (estimate) of the mapping f-1 we can correct the temperature measurement.' Either remove the brackets or extend them: '(an estimate of)'. More fundamentally I haven't fully understood this mapping, any extra explantion would be welcome. From Figure 1 I think that larger corrections are needed at lower airspeeds - is this correct (and true of other aircraft)?*
40  » Done: we added some text
     *line 190,191 'Radiosondes are generally launched at the main hours (00, 06, 12, 18 UTC), as required by WMO, with the majority of launches around 00 and 12 UTC (these timestamps represent the observation at a level of 500 hPa at the whole hour)' 'before the main hours' (often about 45 minutes before, but different NMSs vary). I have heard it said that they should reach 100 hPa at about the main hour. BUFR radiosonde reports have the time of each individual level.*
45  » Done: we added some words
     *line 214 'The most left panel' - 'The left-most panel'*

» Done

*line 215,217 'The reason for the difference in bias with the time of day is not understood. Assuming that the AMDAR bias is constant we observe that the radiosonde bias, changes over the day from overestimation at 06 UTC to neutral at 12 UTC and underestimation at 18 UTC to slightly underestimation at 00 UTC.' Radiosondes are mainly available at 00 and 12 UTC as already stated. Apart from low levels the sample at 00 UTC is quite small - because there are fewer flights at night. The proportion of cargo flights at night may well be higher? Given the sampling issues for both aircraft and radiosondes I would advise against suggesting a diurnal cycle in radiosonde biases. Radiosondes have larger uncertainty in the -stratosphere- in sunlight (Dirksen et al, 2014).*

» Done: we added some words

*line 218 'Figure 4' - interesting variation with aircraft type Looking at the ECMWF bias corrections by aircraft type I see even larger biases for some B787 aircraft (US-AMDAR) and small negative (cool) biases for some Airbus aircraft. Please increase the size of the text labels in figure 4 to improve clarity.*

» Done

*line 225 'the second is an accuracy related correction.' perhaps 'the second comes from the interconnected nature of aircraft measurements: there is a Mach number correction to the temperature and a temperature correction to the Mach number and it appears that avionics systems do not iterate to convergence.'*

» Done

*line 236,237 'The Mode-S EHS information can be applied to correct the AMDAR temperature bias, for those air spaces where Mode-S EHS information is available.' This is not a long-term solution. The meteorological community needs to persuade the aviation industry to improve their avionics/measurements.*

» Done

*line 258 'Painting, J. D.: WMO AMDAR Reference Manual, WMO-no.958, WMO, Geneva, http://www.wmo.int, 2003.' http://www.wmo.int no longer exists and WMO regard this document as superseded, see https://community.wmo.int/activity-areas/aircraft-based-observations/resources/manuals-and-guides*